# CRISPR-Cas9 Mediated Knockout of *SagD* Gene for Overexpression of Streptokinase in *Streptococcus equisimilis*

**DOI:** 10.3390/microorganisms10030635

**Published:** 2022-03-17

**Authors:** Armi M. Chaudhari, Sachin Vyas, Vijai Singh, Amrutlal Patel, Chaitanya Joshi, Madhvi N. Joshi

**Affiliations:** 1Gujarat Biotechnology Research Centre (GBRC), Department of Science and Technology, MS Building, 6th Floor, Sector 11, Gandhinagar 382011, Gujarat, India; armichaudhary97@gmail.com (A.M.C.); sachinvyas13@gmail.com (S.V.); jd2@gbrc.res.in (A.P.); director.gbrc@gmail.com (C.J.); 2Department of Biosciences, School of Science, Indrashil University, Rajpur, Mehsana 382715, Gujarat, India; vijaisingh15@gmail.com

**Keywords:** FasX, SagD, CRISPR-Cas9, knockout, streptokinase, therapy

## Abstract

Streptokinase is an enzyme that can break down the blood clots in some cases of myocardial infarction (heart attack), pulmonary embolism, and arterial thromboembolism. Demand for streptokinase is higher globally than production due to increased incidences of various heart conditions. The main source of streptokinase is various strains of *Streptococci*. Expression of streptokinase in native strain *Streptococcus equisimilis* is limited due to the *SagD* gene-mediated post-translational modification of streptolysin, an inhibitor of streptokinase expression through the degradation of FasX small RNA (through CoV/RS), which stabilizes streptokinase mRNA. In order to improve the stability of mRNA and increase the expression of streptokinase, which is inhibited by SagA, we used CRISPR-Cas9 to successfully knockout the *SagD* gene and observed a 13.58-fold increased expression of streptokinase at the transcript level and 1.48-fold higher expression at the protein level in the mutant strain compared to wild type. We have demonstrated the successful gene knockout of *SagD* using CRISPR-Cas9 in *S. equisimilis,* where an engineered strain can be further used for overexpression of streptokinase for therapeutic applications.

## 1. Introduction

Streptokinase is a thrombolytic enzyme with a molecular weight of 47 kDa that is spontaneously generated by the alpha hemolytic group of *Streptococcus* species. Streptokinase forms a 1:1 streptokinase:plasminogen complex, which is utilized in healthcare to break up blood clots (due to myocardial infractions) and save lives. Streptokinase is used in the treatment of myocardial infections, arteriovenous cannula occlusion, embolism, and deep vein thrombosis. It is worth USD 40 million on the open market. Streptokinase production in other host factors, such as *Escherichia coli,* has a problem with product toxicity or other factors, such as the absence of short regulatory RNAs in the expression host for the stability of streptokinase mRNA transcript. [1,2]. As a result, there is an urgent need to create natural producers of streptokinase. *Streptococcus* (Group A) possesses two component regulatory systems [3], *fasBCAX* and *cov/RS* [4]. Small regulatory ncRNAs (non-coding RNAs) from *fasX* sRNA stabilize the mRNA transcript of *ska* that enhances the streptokinase production [2,5]. *fasBCAX* enhances the streptokinase production and also down-regulates the streptolysin production; simultaneously, *cov/RS* up-regulates the streptolysin production and down-regulates the streptokinase expression [2,4,6]. Small regulatory RNAs generated by fasBCAX stabilize the 5′UTR region of the *skc* mRNA transcript, preventing RNAses from attacking it, resulting in improved streptokinase mRNA transcript synthesis and hence increased streptokinase expression. Under control of *CoV-Rs*, streptolysin is produced by operon *SagABCDEFGHI. SagABCDEFGHI* contains nine total genes for the expression of fully functional streptolysin. The protein backbone of streptolysin is produced by *SagA,* where the 2.7 kDa peptide (SagA) undergoes post-translational modification by the SagBCD complex, which is similar to *E. coli* mcbBCD: B-Cyclohydratase, C-Dehydrogenase, and D-docking proteins, and is transported by SagEFG-mediated ABC transporters [7,8]. Steiner and Malke hypothesized that the knockout of streptolysin-producing genes, which is under the regulation of Cov/RS, may enhance mRNA stability for the *ska* gene and also improve streptokinase production [4]. The direct target SagA (which forms the streptolysin peptide) knockout was not performed as it is involved in cell survival and pathogenicity [9], rather another important factor *SagD* gene was targeted for knockout to abrupt the streptolysin (SagA) post-translation modification, which may lead to enhanced streptokinase production and decreases *Streptococcus* virulence [10,11]. Because SagD is required for complex formation in the SagBCD complex for post-translational modification, changes in the functional protein sequence may obstruct SagBCD complex formation and, as a result, SagA post-translational modification may be halted. In order to knockout the *SagD* gene, we used the CRISPR-Cas9 system from a pCRISPomyces-2 plasmid [12], which is widely used for targeted genome editing of several organisms under control of the *rpsl* promoter system [13,14,15,16,17,18]. The goal of this study was to boost streptokinase expression in *S. equisimilis* by using CRISPR-Cas9 to knockout the *SagD* gene.

## 2. Results

### 2.1. Streptolysin Knockout Strategy Using -PCRISPomyces-2

Streptolysin is thought to act as a negative regulator because it causes the mRNA transcript of the therapeutically relevant streptokinase to become unstable. Streptolysin S knockout using CRISPR-Cas9 genome editing is employed in *S. equisimilis* to increase streptokinase expression. The pCrispomyces-2 single plasmid system was used, together with cas9 and a guide RNA cassette (Figure 1A). To achieve streptolysin depletion, the *SagD* gene knockout was targeted from the *SagABCDEFGHI* cassette in order to disrupt SagA post-translational modification (Figure 1B). Using Golden Gate assembly, a *SagD*-specific guide-RNA sequence-GR3 was inserted, resulting in a recombinant plasmid lacking the *lacZ* gene in order to achieve blue-white selection (Appendix A). Sanger sequencing was also used for confirming the gRNA insertion. The chromatogram for the insertion of gRNA-GR3 into the pCRISPomyces-2 plasmid using Sanger sequencing is shown in Figure 1C,D. For the expression of Cas9 and guide RNA cassette under the *rpsl* promoter system, *S. equisimilis* clones with chimeric pCRISPomyces-2-GR3 were cultured for 2–3 days at 28 °C in TSB medium with 50 mg/L apramycin. Sanger sequencing (genotypic), altered hemolysis activity (phenotypic), and changes in gene expression patterns were all investigated in these clones. The process for identifying knockout clones is briefly discussed in Appendix A.

A graphical illustration of the *SagD* gene editing is shown in Figure 1E,F. The GR3 gRNA was used successfully to knockout the *SagD* gene with a 21-nucleotide deletion (Figure 2A). A mutant Clone of *S. equisimilis* with a 21-bp deletion was further used to examine the expression profiles of both the *SagD* and *skc* genes. According to sequence alignment, these 21 bp deletions/7 amino acid modifications affected the complex formation of the SagBCD complex, which may influence SagA post-translational modification (Figure 2B). To conform this, computational modeling was performed to evaluate the effect of the deleted amino-acids in the mutant. The SagBCD complex is superimposed with *E.coli* MCB complex with significant alpha and beta helices matches (30%) among both (Appendix A). Figure 2C shows that in the wild-type SagBCD complex, seven amino acids form a beta-sheet, which is missing in the mutant. Complete elimination of beta-sheet alters protein folding in relation to other amino acids. A total of seven amino-acid intramolecular interactions revealed that they are involved in the formation of eight strong hydrogen bonds with the surrounding amino acids (Figure 2D, shown in magenta dashes). Glu-745 is an essential amino acid that interacts with three other amino acids, Ala-747, Leu-748, and Ile-749, to maintain the protein structure through sufficient loop formation (Figure 2D). The difference in RMSD between wild-type and mutant SagBCD complexes was 3.36 Å, which is substantial in terms of quaternary protein structure. When the mutant-SagBCD complex was compared to the wild-type complex, significant changes in the beta-sheets were observed, as shown in boxes a and b (Figure 2E). Alpha fold was also used to confirm the protein’s structural changes (Appendix A). During the transformant’s screening, clones with an exact 21 bp deletion were discovered, but no other deletions were discovered due to the uniqueness of the GR3 sequence, which contains the same 21 complementary nucleotides. The original study of Pcrispomyces-2 supports this, as Cobb et al. [12] discovered a deletion in the exact matching sequence to Guide RNAs. Genomics and computational screening revealed the possibility of an altered SagBCD complex, which must be tested in conjunction with our hypothesis of altered streptokinase expression.

After validating genotypic change, phenotypic change was also corroborated using enzyme activity and altered hemolytic activity. Blood agar plating is an appropriate assay for assessing SagA hemolytic activity, whereas fibrin agar plate, clot-lysis, and standard IU/mL assays were used to measure changes in streptokinase production. Figure 3A,B reveals that seven amino acid deletion mutant clones have significantly lowered hemolytic activity (5.2-fold) compared to control/wild-type. Figure 3C showed the clot lysis and fibrin-agar plate assay for comparing streptokinase activity among wild-type and mutant strains. The wild-type zone of fibrin lysis was 7 mm, whereas the mutant strain had an enhanced zone of fibrin lysis of up to 18 mm (Figure 3C). This qualitative change was further validated with quantitative clot lysis activity and hemolytic assay for streptokinase and streptolysin, respectively. Blood is a common factor used in the quantification for both enzymes; however, the substrate for each was different. It is important to separate individual substrates for both enzymes to remove any overlapping activity (Appendix A. Plasma and blood fractions were separated, as described in Appendix A. Plasma contains plasmin, which is a substrate for streptokinase, and erythrocytes, which is a substrate for streptolysin (RBCs). The standard for IU was plotted with the help of STPase available in market for streptokinase clot-lysis activity. To check whether these mutations have affected bacterial viability; a growth curve was performed where we found no significant change in the growth pattern of wild-type strain from mutants ones (Figure 3D). Figure 3E shows that, following mutation, the percentage of clot lysis rose from 44.10 ± 1.01% to 66.69 ± 1.59% (R^2^ = 0.985, *p* (0.0075) < 0.05). The interruption of the *SagD* gene increases the percentage of clot lysis via an increased expression of streptokinase. Wild-type strain showed 4872 ± 111.6 IU, and the mutant showed 7257 ± 175.6 IU (*p* < 0.05) in Figure 3F by comparing clot lysis with the standard. Streptolysin activity appears to be greatly reduced in comparison to streptokinase, which is confirmed using percent heme release (Figure 3G,H). Percent heme release in the wild-type strain is 84.18 ± 1.425, and mutant strain is 16.19 ± 1.068. Changes in protein activity due to *sagD* gene knockout is further given in Table 1. The reduced amount of heme released in the mutant was feasible due to altered streptolysin activity.

### 2.2. Change in the Expression Profile of Streptokinase and Streptolysin

Streptokinase enzyme activity increased considerably after the deletion of a part of the *SagD* gene. However, the expression may be owing to the improved stability of a streptokinase-specific mRNA transcript (4). To validate the enhanced streptokinase expression, real-time PCR was carried out. The change in the expression patterns of both factors *(SagD* and *ska)* was validated using enzyme activity, which was then validated by RT-PCR and SDS-PAGE for protein expression. As mentioned in Section 4.7, RNA isolation and cDNA synthesis were prepared, and an equivalent concentration of cDNA was used to perform the gene expression assay. The study of fold change in gene expression data was analyzed using the 2^−∆∆CT^ method for relative quantification (RQ) in ABI PRISM^®^ 7500 Sequence Detection System Software. Gene expression values were normalized with DNA gyrase endogenous control (control sample RQ = 1). The expression profile of the *SagD* gene was similar in both wild-type and mutant strains (*t*-test, *p* > 0.05). To quantify the further alteration in streptokinase expression, the wild-type showed an RQ = 1.03, while the mutant showed an RQ = 13.58 (*t*-test, *p*-value = 0.0009) (Figure 4A). Streptokinase expression, on the other hand, was dramatically boosted in the mutant strain. According to E.R Gibney et al. [19], a 21-bp deletion may have no effect on the gene expression profile of the *SagD* gene. The transcription of templates with the deletion was not reduced by RNA polymerase. However, protein derived from deletion may lose functionality. Streptolysin S protein with seven amino acid deletion is no longer able to repress streptokinase expression, resulting in streptokinase mRNA remaining stable in the cell for a longer period of time.

### 2.3. Change in Streptokinase Protein Expression Profile

For validating changes in protein expression profiles, both strains were cultivated overnight, and 0.05 OD at 600 nm was adjusted before the final protein expression experiment. Protein was concentrated with ammonium sulfate, and an equal amount of protein was injected into each well after protein estimation with the Bradford test. Protein expression studies of wild-type and mutant strains revealed that the mutant strains had higher levels of streptokinase expression. Protein expression difference in wild-type and mutant strains is shown in Figure 4B left. Lanes 2 and 3 showed increased expression of a protein band, which could represent streptokinase from a mutant strain with a molecular weight of ~47 kDa. In Figure 4B right, ammonium sulfate fraction purification was performed, where the mutant clone MASCOT has a higher streptokinase fraction. In fraction purification, some undesired protein bands were also observed, which can be further removed with higher fraction purification. It can be validated further using Western blotting, although this was not achievable due to a lack of protein antibodies in our lab. To overcome this limitation, an alternative zymography was used. In western blotting, we used protein-specific antibodies; similarly, in zymography, we used a specific substrate for zymography, which is fibrin and thrombin. The difference of clot lysis (of mutant and wild-type) is clearly shown in fibrin zymography. As a result, these protein bands were subjected to MALDI-based peptide mass fingerprinting (PMF) to establish their identity. Peaks corresponding to streptokinase were indicated in red in Figure 4C. MASCOT analysis revealed streptokinase from *S. equisimilis* as the first hit, with query coverage of 48% for both wild-type and mutant samples (Figure 4D). These findings of higher streptokinase expression at the protein level entirely supported the increased streptokinase expression seen in RT-PCR. Streptokinase was also confirmed using zymography with fibrin and thrombin as substrates. After Coomassie staining, clear zones were found, indicating that the fibrin clot was degraded by streptokinase (Figure 4E). Lane 1 of Figure 4E revealed a less distinct zone of fibrin breakdown than mutant in Lane 2. As a result, the enhanced expression profile of streptokinase was confirmed using the aforementioned standard methods. Altogether, we successfully employed a CRISPR-Cas9-based gene-editing technique for increasing streptokinase expression in *S. equisimilis*.

## 3. Discussion

CRISPR-based approaches are widely employed in biotechnology and medicine, ranging from basic to applied research. We used CRISPR-Cas9 genome editing in a prokaryotic bacterium to increase the production of clinically important streptokinase. Streptokinase is an enzyme that is used for the treatment of myocardial infections, arteriovenous cannula occlusion, embolism, and deep vein thrombosis. It is worth USD 40 million on the open market. Several techniques for boosting streptokinase expression in wild-type *S. equisimilis* strains with various growth agents [20], plasmid-based genetically engineered *E. coli* strain [21,22], and eukaryotic expression yeast [23,24] were already available. Although the maximum achieved production was 720 mg/L streptokinase using *E. coli BL21[9DE3]* strain [22]. In the present study, we developed a model system in *S. equisimilis* using CRISPR-Cas9-mediated knockout of streptolysin to enhance the production of streptokinase. According to E. Charpentier and B. Kreikemeyer et al. [2,5], streptolysin is a negative regulator of the expression of streptokinase. Streptolysin is produced under the control of the CoVRS/S system, which plays a key role in the expression of virulence factors. *Streptococcus* possess small FasX RNA, which imparts stability to streptokinase mRNA transcripts and protects from ribonucleases [2]. We targeted the *SagD* gene, which forms a complex with SagBC to perform the post-translational modification of SagA to alter/knock down streptolysin S production.

The *SagD* gene was mutated using pCRISPomyces-2 + gRNA. Protospacer specificity till 15–17 nucleotides is necessary for the efficient binding to a target without any off-target effect. In pCripomyces-2, Cas9 is expressed under the control of the *rpsl* promoter, which is 30S ribosomal protein S12, and gRNA is expressed under the *GAPDH* (glyceraldehyde-3-phosphate dehydrogenase) promoter. Both promoters have been demonstrated to be effective in a wide range of microorganisms, including bacteria, fungi, algae, and protozoa [12]. BLASTn was performed while designing the gRNA sequences to avoid any off-target effect. The sequences with minimum off-target effects were selected. A total of four protospacers were utilized, and GR3 was successful in editing the *SagD* gene. As a result of the *SagD* gene mutation, streptolysin S production was reduced, which was confirmed using a hemolysis assay. We discovered a considerable reduction in hemolytic activity. (Figure 3). According to Vi.Nizet et al. [25], the whole *SagABCDEFGHI* cassette is important for the expression, post-translational modification, and transport of streptolysin. The deletion of any of the *SagABCDEFGHI* genes can alter the SagA function as streptolysin. The change/loss in amino-acid sequence in protein might lead to structural and functional disruption [26,27]. After streptolysin alteration, a stable mRNA transcript of streptokinase was formed in the cell, and the mRNA pool-specific of streptokinase was also raised. SDS-PAGE and RT-PCR were used to confirm an increase in streptokinase transcript. The gene expression profile of streptokinase has significantly increased by 13.58 RQ (*p*-value < 0.05). This result supports the hypothesis showcasing stability of streptokinase mRNA transcripts leading to the prolonged expression of streptokinase in *S. equisimilis*. SDS-PAGE, in Figure 4B, clearly indicates the enhanced expression of streptokinase (47 kDa). To validate the SDS-PAGE expression profile, a protein band of ~47 kDa was excised and subjected to PMF-MALDI, with results hit-specific to streptokinase with more than 48% coverage in a MASCOT search, further confirming the increased streptokinase expression at the protein level.

Studies on the expression of streptokinase with different expression vectors/strategies and the host are available. Some have used *Streptococcus agalactiae* with nonspecific ethyl methanesulfonate (EMS)-based chemical mutagenesis [28], *Streptococcus sanguis*-produced lowered-molecular mass-containing streptokinase [29], and *B. sublitis* with C-terminally processed streptokinase [30]. A relative comparison of streptokinase activity in different hosts was as follows, *Saccharomyces pombe* (2450 IU), *Streptococcus equisimilis* (100–150 IU), *Pichia pastoris* (3200 IU), and *E. coli* (1000–1500 IU) [28,31,32,33]. All these studies observed the degradation of mature streptokinase 47 kDa to inactivated 44 kDa [29,30,34]. This 44 kDa inactive protein might be a degraded/truncated product of unstable protein, which is not observed in our case. In this present study, we developed a model system for streptokinase expression without requirements for the inducer, antibiotic, and optimized media in the native strain. Our modified strain can generate streptokinase at a lower cost. Another advantage is that streptokinase degradation was not found in the native strain. In terms of IU per ml, the fold increase in streptokinase-specific activity was 1.12. The fold change in streptokinase enzyme activity between wild-type and mutant was 1.48, with a *p*-value less than 0.05. Despite considerable mRNA transcript expression for streptokinase, its enzyme activity is only 1.48 times higher. This could be due to the fact that we measured enzyme activity in crude supernatant (protein is not purified) and streptokinase’s instability. Our primary aim of streptokinase mRNA transcript has been met; however, we have seen a significant increase in protein expression (*p* < 0.05), though it is not significant when compared to mRNA transcript expression. Protein stability could be one of the causes behind this. When performing protein expression assays, we must assess protein activity on the same day, or we will get degraded protein the next day. Many scientists are concerned about the stability of streptokinase (in terms of protein degradation), for example, the *Icikinase* injection vial of streptokinase, which has a half-life of only 18 min (https://www.medicines.org.uk/emc/product/4256/smpc#gref till date 25 February 2022). We are employing computational modeling and replica-based studies to identify a stable buffer system where we identified mg^+2^ ion as vital for protein expression. Maintaining the pH and a stable cation-based buffer system while culturing in growth media has been shown to increase streptokinase production and will be used in our future experiments [20].

The CRISPR-Cas9 system enabled us to alter the *SagD* gene, which is an alternative approach for enhanced expression of streptokinase. Here, the native strain can be used for industrial streptokinase production, where its virulence is altered, and the desired product expression is increased. No inducer is required, as the product is generated by native strains only. CRISPR-based gene editing can be useful for the production of efficient microorganism models for the production of industrially important products.

## 4. Materials and Methods

### 4.1. Microorganisms, Plasmids, and Reagents

The *S. equisimilis* culture used in this study was obtained from ATCC (43079). *Escherichia coli* Top10F’ procured from Invitrogen, Thermo-Fischer Scientific, Waltham, MA, USA [35] was used for routine experimental work and also used for the preparation of competent cells and cloning of guide RNAs and propagation of plasmids. The pCRISPomyces-2 vector used in this study was obtained from Addgene, Watertown, MA, USA (Code: 617374). All the media and chemicals were used in this study were purchased from Hi-media (Mumbai, India) and Thermo-Fischer Scientific (Ahmedabad, India). All the clones and cultures were stored in (25% *v*/*v*) glycerol at −80 °C for long-term storage and further study. For designing primers listed in this manuscript, GenBank assembly accession number GCA_900474875.1 genome sequence was used. In Appendix A, a list of different clones used in this study is given. A list of primer sequence and ID is given in Appendix A. The CRISPR-cas9 image in Figure 1F is taken from the wiki-CRISPR-gene editing webpage.

### 4.2. Media and Culture Conditions

*S. equisimilis* was cultivated in Trypton soya broth (TSB) medium (HiMedia, Mumbai, India). *E. coli* Top10F’ strain was cultivated in Luria-Bertani (LB) medium (HiMedia, Mumbai, India) with an appropriate concentration of antibiotic (Apramycin) 50 mg/L. The two clones obtained have similar genome editing, and one particular clone is taken into the study.

### 4.3. Design and Construction of sgRNAs Plasmids

The protospacer sequence {~20 nucleotides (nt)}, a complementary sequence targeting the *SagD* gene in *S. equisimilis* to construct the sgRNA design was performed using the available online tool, CRISPOR (http://crispor.tefor.net/ till date 25 February 2022) [36]. The genome of *S. equisimilis* was used by providing the accession id to the Tefor team (from webserver CRISPOR.org) to generate efficient gRNA sequences with minimum off-target effect. A 1693-bp-long target sequence for the *SagD* gene was used as an input. All of the 157 possible gRNAs sequences containing PAM (Protospacer Adjacent Motif) sequences were analyzed based on efficiency and off-targets for mismatches. Four possible 24-nucleotide-long protospacer sequences (4 nt 5′ sticky end + 20 nt spacer sequence) with sticky-end ACGC on the forward sequence and AAAC on the reverse sequence were chemically synthesized [37]. The protospacer sequence was generated by equimolar ligation as previously reported (Appendix A) [37]. The gRNA was inserted via sticky-end cloning into pCRISPomyces-2 plasmids based on the Golden-Gate assembly protocol using *BbsI* (type IIs restriction enzyme) (Appendix A) [12].

### 4.4. Transformation of Constructed Plasmid into E. coli Top 10F’ and S. equimilus

The irreversible reaction containing only intact plasmids with the correct protospacer insert at the end was used for the transformation into chemically competent *E. coli* TOP10F’ (~10–100 ng) using a heat-shock method [12,38]. It was spread on LB agar containing apramycin (50 µg/mL) pre-coated with appropriate concentrations of IPTG (1 mM) and X-Gal (40 µg/mL) for blue-white colony screening [39]. The white colonies observed were screened for plasmid isolation and confirmation by PCR using previously designed primers [12]. The protospacer insertion was also confirmed using Sanger sequencing.

The recombinant plasmid was first screened using PCR, where pCRISPomyces-2 plasmid having guide-RNA will show in a ~385 bp band (transformants-white colonies), and those having intact plasmid will show larger fragments of ~800 bp for wild-type plasmids (non-transformants-blue colonies). The recombinant plasmids were prepared with different gRNA cassettes (Appendix A), out of which GR3 was successfully used for the knockout of streptolysin. pCRISPomyces-2-GR3 was maintained in *E. coli top10F’* before being electroporated into *S. equisimilis* (Appendix A).

Recombinant plasmids with sgRNA were transformed into electrocompetent *S. equisimilis* using the electroporation unit of a Bio-Rad X gene pulser. The electrocompetent cell’s preparation was performed according to the previously reported protocol [40]. Briefly, ~0.4 optical density (O.D.), cells were previously grown in glycine medium, were washed four times with pre-chilled 0.5 M sucrose. The cells were suspended in 250 μL of a 10% (*v*/*v*) glycerol and 0.5 M sucrose solution and divided into aliquots and stored at −80 °C for future electroporation experiments. The electroporation conditions followed were 25 kV/cm, 200 Ω, and 25 μF [41]. Following the electroporation experiment, the cells were grown in 1 mL of TSB broth medium at 30 °C for 1 h. The transformed cells were then plated on a TSB medium containing apramycin at a concentration of 50 µg/mL and incubated at 28 °C for 72 h.

### 4.5. Screening of Knocked out Clones Using Sequencing and Hemolysis Activity

Each of the colonies observed was tested for the presence of recombinant plasmid using the vector-specific forward primer 5′ACGGCTGCCAGATAAGGCTT3′ and reverse primer 5′ TTCGCCACCTCTGACTTGAG3. The positive colonies were further grown in 5 mL of 50 µg/mL apramycin containing a TSB medium at 28 °C and 120 rpm. The positive colonies were screened for DNA sequencing of the PCR product by *SagD* gene-specific primers for possible modifications and knockout (Appendix A). Detailed screening of obtained clones was explained in Appendix A. Mutant-GR3 (Clone 7) with 21 bp deletion was used in further research.

### 4.6. Real-Time PCR for Gene-Expression Profiles

Total RNA extraction was performed from the wild-type and mutant clones using the RNeasy plus mini kit (QIAGEN). RNA was quantified by qubit 4. The cDNA synthesis was performed using the QunatiTect Reverse transcription kit QIAGEN [42]. The cDNA concentration was quantified for both wild-type and mutant samples, such that one reaction has 100 ng cDNA. RT-PCR (Fast 7500 of Applied Biosystems) was performed using TB-Green^TM^ Premix Ex Taq^TM^ II. *SagD* gene expression was checked with forward primer *SagD_*RT*_F* 5′ GACAGCCTCTCATACAACAC 3′ and reverse primer *SagD_*RT_R 5′AGCGGATTATCCTCTCCAAC 3′, and *ska* gene expression was checked using forward primer *SK*_RT_F 5′ ATACATCTTGACGGGTCAGG 3′ and reverse primer *SK*_RT_R 5′ AAGAGACCCTGCTGCCAT 3′. DNA gyrase was used as an endogenous control, and *DNA*_G_F’ 5′ GCGAACAATATGCTCATGGACC 3′ and *DNA*_G_R’ 5′ CTTATGAGACTGGTAAGGGA 3′were used for the same. In the reaction step, the 1st stage was the holding stage at 52 °C for 2 min, initial denaturation at 95 °C for 2 min, and in the cycling stage (40 cycles), the 1st step was at 95 °C for 30 s and an annealing temperature of 58 °C for 30 s followed by the melting curve stage [43]. Then, expression profiles of both *SagD* and *Ska* genes were analyzed using RT-PCR, and a comparison was made by the 2^−ddCt^ method.

### 4.7. MALDI-TOF/MS for Confirmation of Streptokinase Expression on SDS-PAGE

SDS-PAGE was performed using crude extract supernatant from cultures overnight grown for both wild-type and mutant using the prescribed protocol in Sambrook [44]. For the confirmation of proteins, the prominent expected bands for Streptokinase (Ska) were eluted from the gel by cutting and subjected to in-gel trypsin digestion [45]. Eluted peptides were used for MALDI-TOF/MS analysis in an Auto flex speed MALDI/TOF/TOF spectrophotometer (Bruker Daltonics, Germany). The output data were submitted to the MASCOT (MS/MS ion search engine from Matrix science) server (http://www.matrixscience.com/server.html till date 25 February 2022) for peptide mass finger-printing analysis.

### 4.8. Zymography

The presence of the enzyme streptokinase was confirmed via zymography [46]. In 10 mL of resolving gel, 0.12 percent (*w*/*v*) fibrinogen and 1 unit/mL thrombin were added. A layer of stacking gel was layered, and a quantified protein solution was placed into each well. Coomassie blue staining was performed after SDS-PAGE electrophoresis at 4 °C to observe the zone of clot lysis.

### 4.9. Streptokinase Blood Clot Lysis and Fibrin Plate Assay

Crude enzyme supernatant was obtained from the culture of both wild-type and mutant strains and was subjected to clot lysis assay [47]. Wild-type and mutant strains were grown in TSB media overnight (12 h), and on the next day, equal O.D. was set to 0.05 at A_600nm_ to maintain an equal number of cells in each experiment. In brief, 10 mL of fresh blood sample was collected from a voluntary healthy individual (as per the Institutional Animal Ethical Guideline). Further, 500 µL of blood was collected and allowed to clot for 90 min in 1.5 mL micro-centrifuge tubes. The clots were centrifuged at 4000 rpm for 5 min at 4 °C to avoid an excess amount of unclotted/liquid section. The enzyme activity was estimated by a change in the clot weight (initial vs. final) by incubating it with an enzyme for 90 min. The standard was plotted with available streptokinase *STPase* from Cadila Pharmaceuticals Ltd., Ahmedabad, India, with a 1,500,000 IU (International Unit). The IU of streptokinase was measured as the ability to cause the lysis of a fibrin clot via the plasmin system in vitro. The standard was diluted such that 100 µL contains 3000 IU units (10 mg). A total of 8 sets of 1.5 mL micro centrifuge tubes containing clotted blood were taken. Standard STPase dilutions from 1000–7000 IU were subjected to clot lysis in 7 micro-centrifuge tubes and 1 tube with PBS for negative control. After 90 min, the clot lysed blood was centrifuged at 4000 rpm for 5 min at 4 °C to weigh the remaining clot. The initial clot weight was annotated as *W*1, and after treatment, it was annotated as *W*2. To calculate the percent (%) of clot lysis, the following equation was used.
$$\frac{W1-W2\times 100}{W1}$$

The standard was plotted for percent clot analysis vs. IU. For two experimental samples, wild-type and mutant derived proteins, ammonium sulfate concentrated solution with an equal amount of protein 0.65 mg were subjected to clot lysis. Based on the percent clot lysis, their IU and specific activity were determined.

Fibrinolytic activity was estimated using a protocol described earlier [48]. A total of 100 µL of thrombin (10 NIH U/mL in 50 mM pH 7.2 sodium phosphate buffer) and fibrinogen from human plasma (5 mg) was dissolved into 5 mL of 50 mM pH 7.2 sodium phosphate buffer. The thrombin and fibrin solutions were mixed into 2% agarose, which was dissolved into 50 mM pH 7.2 sodium phosphate buffer and poured into a Petri dish. The Petri dish was kept at 30 °C to form a fibrin clot layer. A 5 mm well was created using a cup borer. The protein solution was added and allowed to clot by incubating the plate at 37 °C for 12 h. Plasmin (1000 IU) was used as a positive control. The zone of fibrin lysis was measured in mm.

### 4.10. Streptolysin Heme Release Assay

The *Streptococcus* species contains two types of streptolysins, streptolysin S, which is oxygen sensitive, and streptolysin O, which is oxygen stable [49]. We used *S. equisimilis* containing the streptolysin S. The streptolysin activity was measured by percent release of heme during RBC lysis [50]. The substrate for streptolysin S is human erythrocytes. A total of 10 mL of blood was collected from a voluntary healthy individual (as per the Institutional Ethical Guideline). The blood was centrifuged at 3000 rpm for 30 min at 4 °C. As shown in the Appendix A, the complete erythrocytes and plasma section were separated at 4 °C. Erythrocytes were washed in 10 mL PBS three times to remove impurities from the blood. Erythrocytes were dissolved in 10 mL PBS.1:20 diluted erythrocytes solution was used for studying streptolysin activity using percent (%) heme release at A_540nm_. In a 1.5 mL micro centrifuge tube, the solution of erythrocytes was placed and ammonium sulfate was used to precipitate the supernatant with equal amounts of protein, 0.65 mg. The tubes were incubated for different time intervals, ranging from 10 to 120 min. On completion of each time interval, the tubes were centrifuged at 1000 rpm to settle down RBCs, and only the lysed RBCs heme section present in the supernatant was determined by measuring OD at A_540nm_.

### 4.11. SagBCD Complex Computational Modeling

The X-ray crystal structure of the SagBCD complex was not determined yet [8]; however, it has shown structural similarity (30%) with the *E. coli* McbBCD complex. Consequently, this complex was built by performing homology modeling using the McbBCD as a template, which is available in the PDB (Protein data bank: 6GRI). We have used homology modeling methods to 3-D model the SagBCD complex using Schrodinger 2021.2 licensed suite. As there is only 30% identity for reference structure, we used the PRIME-STA (Single templet alignment) module from Schrodinger, which is best suited for low sequence identity.

### 4.12. Growth Curve for Wild-Type and Mutant Strains

Wild-type and mutant strains were grown in TSB at 37 °C. The next day, OD = 0.05 was set for both strains, and the absorbance at A600 nm was measured at regular time intervals till the stationary phase [51].

## 5. Conclusions

Streptokinase is an industrially important enzyme and is used as a drug for the breaking down of blood clotting in some cases of myocardial infarction (heart attack), pulmonary embolism, and arterial thromboembolism. In this study, we have successfully designed a CRISPR-Cas9-based system for the *SagD* gene (inhibitor for streptokinase expression) knockout in the wild-type *S. equisimilis* strain. We have found a 13.58-fold increased transcript level of streptokinase in the knocked out strain compared to the wild-type strain. Our CRISPR-Cas9-based approach can be used in other microorganisms for genome editing for other industrially important therapeutic production.

## 6. Highlights

Streptokinase is an important enzyme used as a drug for the breaking down of blood clotting.The *S. equisimilis* strain was used for the expression of streptokinase.The CRISPR-Cas9 system was employed for genome editing of the *SagD* geneWe observed a 13.58-fold higher streptokinase expression in the knocked out strain compared to the wild-type strain, and the fold change in enzyme activity was 1.48 (*t*-test, *p* > 0.05).

**Limitations of this study and future aspects:** The authors of this manuscript created a methodology for increasing protein expression; however, this study looks at conforming protein activity on a preliminary basis based on crude supernatant. Because streptokinase is expressed extracellularly and this approach was created in a native strain, no tag (for example, histag) is available for the separation of this protein. We are developing a protein purification procedure in native strains employing ammonium sulfate precipitation and subsequent chromatography-based applications. We are also attempting to alter this protein in order to make it more stable, which may expand its therapeutic potential. We are employing computational modeling and replica-based studies to identify a stable buffer system where we identified the mg^+2^ ion vital for protein expression. One of our two techniques for finishing this research project is to use the C-terminal portion of streptokinase, which has a higher sensitivity complement system in humans. We want to mutate this C-terminal region in ways that minimize streptokinase reactivity during treatment. The second is to develop a chromatography-based purification process for streptokinase purification using our own antibodies. For protein purification, we are also fattempting to insert histag into the native strain for ni-NTA purification.

## Figures and Tables

**Figure 1 microorganisms-10-00635-f001:**
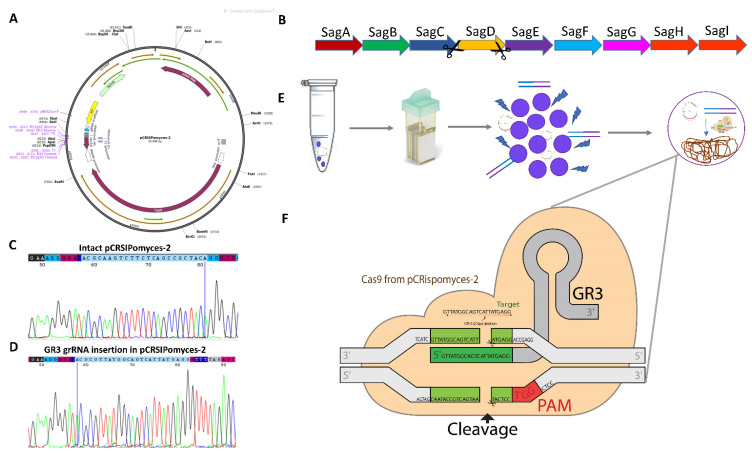
CRISPR-Cas9 system for knockout. (**A**) Intact PCRISPomyces-2 plasmid. (**B**) Streptolysin cassette showing *Sag-ABCDEFGHI* genes for streptolysin biosynthesis. (**C**) Intact/Un-edited pCRISPomyces-2 plasmid. Incorporation of GR2 gRNA sequence into the PCRISPomyces-2 plasmid. (**D**) Incorporation of GR3 gRNA sequence into the PCRISPomyces-2 plasmid. (**E**) Workflow showing an experimental design for inserting the pCRISPomyces-GR3 using electroporation. (**F**) Schematic diagram showing the mechanism for gene editing using GR3.

**Figure 2 microorganisms-10-00635-f002:**
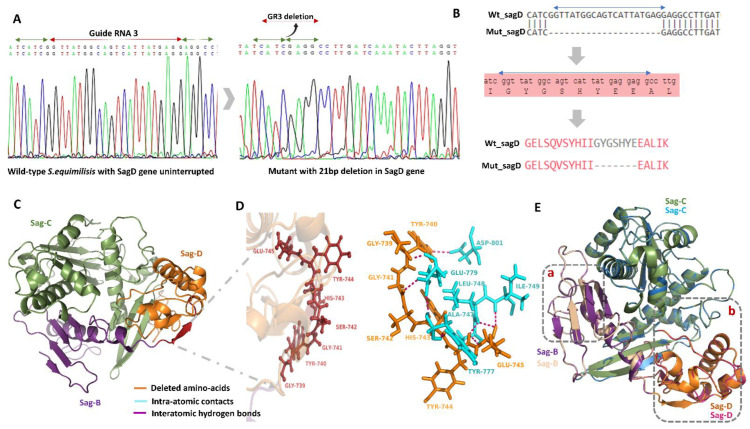
Gene editing of the *SagD* gene using GR3-PCRISPomyces-2. (**A**) Native *SagD* gene without any gene editing and mutant *SagD* genes with 21 deleted nucleotides complementary to GR3. (**B**) Alignment of wild-type and mutant SagD genes (nucleotide sequence) for exploring change in the amino acids sequence. (**C**) SagBCD complex highlighting deleted amino acids with red color. (**D**) The significance of the seven highlighted amino acids in the synthesis and folding of the SagBCD complex. Intra-atomic connections were depicted in magenta using hydrogen bonding within less than 3°A distance (dashed lines). (**E**) Structural superimposition of wild-type and mutant SagBCD complexes. SagB, SagC, and SagD are shown in purple, green, and orange color in wild-type while for mutant, cream, blue, and pink were shown, respectively. The major structural difference is highlighted in boxes a and b. Areas with comparable folds will have merged colors, whilst areas with different folds will not have merged regions.

**Figure 3 microorganisms-10-00635-f003:**
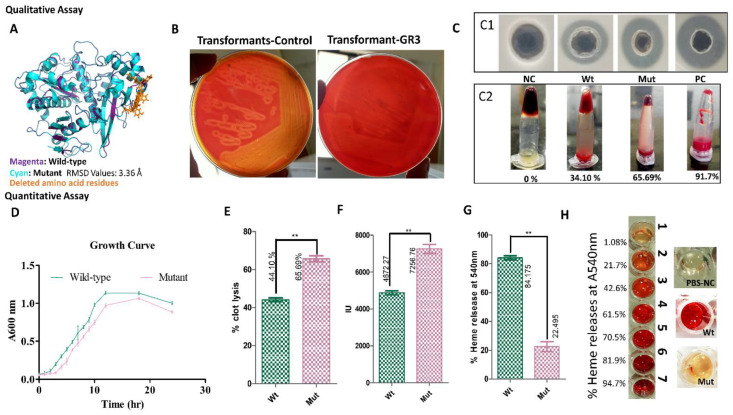
Quantitative and qualitative protein assays of streptokinase and streptolysin. (**A**,**B**) Wild-type and mutant strains were streaked on blood agar plates with equal 0.05 OD_600nm_, to show altered hemolytic activity. (**C**) Streptokinase clot lysis activity: (**C1**) Zone of fibrin lysis by streptokinase in the thrombin-fibrin agar plate; (**C2**) Clot lysis assay for streptokinase. NC is abbreviated as a negative control, Wt (wild-type), and Mut (mutant), and PC is a positive control. (**D**) Growth curve for wild-type and mutant strains. (**E**) Percent clot lysis compared in wild-type and mutant strains (**: represent significant with *p* < 0.05). (**F**) Comparison of IU units within wild-type and mutant strains. (**G**) Streptolysin activity represented using a percentage of heme released by lysis of erythrocytes. (**H**) Change in percentage heme release measured at A_540nm_ for streptolysin.

**Figure 4 microorganisms-10-00635-f004:**
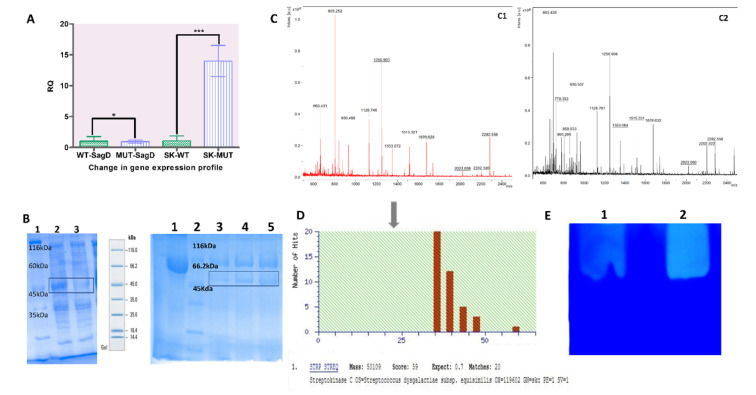
Determination of change in the expression profile of streptokinase. (**A**) Real-time PCR data from the gene expression panel showed the expression of *SagD* and ska genes in wild-type and mutants. The green color bar shows the gene expression data for wild-type, and the purple color bar shows the data for mutant. * represents not significant; *** represent significant with *p* < 0.05. (**B**) SDS-PAGE analysis showed enhanced expression of streptokinase at the expected size ~47 kDa. (**left**) Lane 1: Protein ladder. Lane 2: Mutant GR3 clone. Lane 3: Wild-type with crude 100% acetone precipitated. (**right**) SDS-PAGE showing fraction purification of crude supernatant using ammonium sulfate. Lane 1: BSA. Lane 2: Protein ladder. Lane 3: Wild-type. Lane 4: Mutant with 50% fraction. Lane 5: Mutant with 70% fraction. (**C1**,**C2**): Peak pattern obtained using MALDI matched with streptokinase protein. Red lines in peaks show 48% and match with the streptokinase protein from wild-type, and black with the mutant clone. (**D**) MASCOT analysis also showed streptokinase protein in first hit. (**E**) Fibrin Zymography showing high fibrin-thrombin clot lysis activity for the mutant strain. Lane 1 shows the wild-type strain, and Lane 2 shows the mutant strain.

**Table 1 microorganisms-10-00635-t001:** Summary of streptokinase expression in wild-type and mutant strains of *S. equisimilis*.

Strain	Protein (mg)	Enzyme Activity IU	Specific Activity IU/mg	Percent Hemolytic Change	Fold Change in Gene Expression
Wild-type	**1.78**	4872 ± 111.6 IU	2737.2 ± 88.65 IU/mg	84.18 ± 1.425	1.03
Mutant	**2.35**	7257 ± 175.6 IU	3087.9 ± 105.65 IU/mg	16.19 ± 1.068	13.99

## Data Availability

All data generated are included in this manuscript.

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
