# Peer review of "CRISPR-Cas9 Mediated Knockout of *SagD* Gene for Overexpression of Streptokinase in *Streptococcus equisimilis"

_microorganisms, 2022, doi:10.3390/microorganisms10030635_

Round 1
Reviewer 1 Report
Review comments:
In this manuscript, Armi et.al utilized the CRISPR-Cas9 method to alter sagD gene and characterized its influence to the streptokinase expression level in Streptococcus equisimilis. This is mainly based on the previous study about the regulation of SagD gene toward streptokinase expression. This is an application of CRISPE-Cas9 technology in bacterial genomic editing for possible therapeutic application. I would endorse this paper for publication after following points to be addressed:
Major comments:
- In the result 2.3, the author characterizes the protein expression level between mutant and WT. It’s obvious that the expression of streptokinase owns a big difference from Fig.4B. But the question is that the total protein amount of each sample looks different though the author mentioned Bradford method was used to estimate the total amount. A stable control protein may be applied to normalize the sample loading on SDS-PAGE and then compare the streptokinase expression.
- In fig3E and 3F, the clot lysis induced by the expression increasing of streptokinase various not that much compare with the protein amount difference showed in fig.4B. The author needs to discuss this point.
- The novelty of this paper is CRISPE-Cas9 technology in bacterial genomic editing for possible therapeutic application. As the author state in the limitation part, the purification of the protein is a question to be solved. Why not apply an affinity tag at the terminal of streptokinase with CRISPE-Cas9 method again. I hope the author try to do this or make a discussion about this if they think it’s hard to do.
Minor comments:
- Please make sufficient literature citation in the whole main text. For example: line 42, line 56, line 94 and so on.
- Title of result 2.1. “Streptolysin knockout strategy using -PCRISPomyces-2”. Actually, it is SagD knockdown. In the whole paper, “knockout”, “knockdown” and “alter” was used. SagD is not knockout as just a deletion introduced. Please revise it.
- In figure 2C, please color SagB, SagC and SagD in different color;
- In fig 2D, please label all the interaction amnio acid as mentioned in the main text. A stick model of amino acid without showing hydrogen atoms may be much clearer.
- For the model, a predict model using AlphaFold2 is preferred. The author can compare the current one with it and make the model more reliable. And I would like the author discuss more about why this 7aa deletion influence the SgaA post-translation modification, is this 7aa nearby the enzymatic active sites? Or any other reason?
- Figure 4C, 4D is not clear, please provide high resolution one.
- Figure 4E is not clear and no big difference can be observed, please replace with much clearer one.
Author Response
we have uploaded pdf file as response sheet

Reviewer 2 Report
The authors have done interesting and important work but unfortunately the work is not presented good enough.
This paper needs extensive text and data editing. Introduction is not sufficiently described (for example expression of gene ska, how is Cov/RS regulated, better explain the role of SagD). There are also many typos, different font sizes, italic is missing in many places (genes and organisms should be written in italic), capital letters should be checked as well (in case of a gene, it is not written with the capital letter; Figure is always with the capital letter). It is very difficult to name all typos in the text.
As for Results, in supplementary data Figures are too small and difficult to see. Many data there are unnecessary and only complicate reading and understanding. For example, failed deletion mutants, alignments which are not labelled properly and key differences not emphasized. In Figure S2A - I don't understand why SagC and SagE are labelled? Description in Materials and methods is sufficient and perhaps keep only positive results in supplementary results.
MAJOR comments
Please label samples in the same way for qPCR (genes in italic), also better label results for 4C, 4D and 4E and include controls. Those results are crucial for the whole work and should be presented with all details and controls.
Author Response
We have uploaded pdf file as response sheet

Reviewer 3 Report
The manuscript presents interesting achievements in the area of genome modification. While the manuscript is detailed, some aspects should be addressed.
I as understood, only was mutant with deletion of 21-nucleotides (7-aa) were used in this study. Why other mutants with a frame-shifting were not observed? Was only one mutant characterized or independently obtained mutants were included to account for any biological variation and possible Cas9 off-targets?
While the great increase in the expression was observed, the streptokinase-specific activity was only slightly increased, how can that be explained? The information about the streptokinase-specific activity should be included in the abstract of the manuscript.
Other comments:
L38-L39: remove enter
L61: ‘golden gate’ – capitalize
L66: remove additional comma (Figure 1C-D..)
L66-L68: rephrase the sentence to avoid confusion.
L68: add a comma (“system These”)
L129: “real-time PCR” – name must reflect that it was quantitative (e.g. real-time quantitative PCR)
L148: “600nm” space missing
L149-L172: font issue
Figure 1F: this part was repurposed from Wikipedia page (https://en.wikipedia.org/wiki/CRISPR_gene_editing) – please cite appropriately
OD600nm – make the uniform way of writing throughout the text
L220: improve the way of writing “CRISPRCas9”
L221: improve citation format “E.charpentier and B.Kreikemeyer et al”
Some elements of the Discussion section could be transferred to the Introduction section (e.g. L213-215) or placed in the results sections to present a clear discussion of results. The discussion section could be extended by discussing other aspects, such as growth conditions that could increase streptokinase production in obtained mutants. Also, future prospects for this work could be mentioned.
Author Response
We have uploaded response sheet as a pdf file.

Round 2
Reviewer 1 Report
In this revised manuscript, the author improved a lot. There is still some points need to be improved as I mentioned in the first comment:
1. Previous comments: "In the result 2.3, the author characterizes the protein expression level between mutant and WT. It’s obvious that the expression of streptokinase owns a big difference from Fig.4B. But the question is that the total protein amount of each sample looks different though the author mentioned Bradford method was used to estimate the total amount. A stable control protein may be applied to normalize the sample loading on SDSPAGE and then compare the streptokinase expression."
In the revised version, the author indicate they use BSA as the control. But what I mean is they should find a protein in the total protein extraction and normalize the sample loading amount and compare the streptokinase expression. This should be an endogenous one, just one similar as tublin or GADPH was applied for WB in the cell biology experiment.
2. FIgure 2D, the amino acid are shown is not clearly though stick model was applied this time. Please read other protein structure paper to check how the amino acid are generally shown. Figure 2C was revised while the figure 2E not shown in similar way.
Please check the paper more carefully and revise again.
Author Response
CRISPR-Cas9 mediated knockout of SagD gene for overexpression of streptokinase in Streptococcus equisimilis
Armi M. Chaudhari1, Sachin Vyas1, Amrutlal Patel1, Vijai Singh2, Chaitanya G. Joshi1 and Madhvi Joshi1* Author response sheet
Reviewers are appreciated by authors for conducting critical reviews and flagging crucial topics in order to make our work more publishable. Major changes for second revision is highlighted in red and blue color. Please find attached file for detailed review.
In this revised manuscript, the author improved a lot. There is still some points need to be improved as I mentioned in the first comment:
Reviewer concern’s: 1. Previous comments: "In the result 2.3, the author characterizes the protein expression level between mutant and WT. It’s obvious that the expression of streptokinase owns a big difference from Fig.4B. But the question is that the total protein amount of each sample looks different though the author mentioned Bradford method was used to estimate the total amount. A stable control protein may be applied to normalize the sample loading on SDSPAGE and then compare the streptokinase expression."
In the revised version, the author indicate they use BSA as the control. But what I mean is they should find a protein in the total protein extraction and normalize the sample loading amount and compare the streptokinase expression. This should be an endogenous one, just one similar as tublin or GADPH was applied for WB in the cell biology experiment.
Author’s response: We apologise for any confusion this has caused. It was all about endogenous control in this case. The reviewer is fully correct in stating that any alteration in gene expression, control is essential. To standardize quantitative PCR results for RT pcr, we employed DNA gyrase as an endogenous control. We have shared the raw data for the same here. The RT pcr slot was always in sync with the SDS page. Endogenous control clarification has been added to the material and approach. The same sample was used for the SDS page and the RT PCR, with the initial quantity of bacterial cells kept the same so that the wild type and mutant had the same OD, as well as the incubation duration. We've looked at both rna and protein expression in this case. We have DNA gyrase as an endogenous regulator for rt pcr, which is a housekeeping gene that is widely employed in prokaryotic systems. According to the literature, researchers always run a control and experimental sample with equal protein amounts, but endogenous controls are only retained if western blotting is performed, which is not possible in our scenario.
Figure 1: Melt curve analysis from RT PCR for streptokinase, sagD and endogenous control DNA gyrase.
Figure 2: Induction of protein
Above figure 2 is took from Ariane Leites Larentis et al (https://bmcresnotes.biomedcentral.com/articles/10.1186/1756-0500-7-671) for reference that in case of protein expression profile control and experimental were run as such that same amount of protein is loaded in well and the we check weather desired size of protein is expressed at higher level or not. According to your previous concern that “But the question is that the total protein amount of each sample looks different though the author mentioned Bradford method was used to estimate the total amount” is solved in first page image in figure 3. To kept endogenous control in SDS-Page is possible only if complete proteome profile of page is known if we take DNA gyrase in SDS page for endogenous control we need to know amount of DNA gyrase from crude supernatant, which is rigorous. Having endogenous control in RT-PCR is enough to state higher expression of streptokinase because here we are using DNA gyrase specific probes or primers, as shown in figure 1 of document.
- Figure 2D, the amino acid are shown is not clearly though stick model was applied this time. Please read other protein structure paper to check how the amino acid are generally shown. Figure 2C was revised while the figure 2E not shown in similar way.
Author’s response: please find attached figure with required suggestions. SagB, SagC, and SagD are shown in purple, green, & orange color in wildtype while in mutant cream, blue and pink were shown respectively
Please check the paper more carefully and revise again.
Author’s response: Apart from above concerns, In manuscript protein were annotated as SagBCD (COMPLEX), genes where sagD, skc, etc, and organisms’ names were highlighted in italics for example S. equimilius.

Reviewer 2 Report
I thank the authors on comments and improved manuscript. I find this version much better but there are still a lot of typos unfortunately. Please check that Figures and Tables are always written with the capital letter in the text, genes in italic, proteins with the capital letter, species in italic. Please check lines: 41, 45, 56, 57, 58, 59, 77, 88, 88, and many others ... in materials and methods, also it is SDS-PAGE (line 227
add "from ribonucleases" in line 250
please define acronim MASCOT
in highlights section - sagD gene in italic
C-terminal, not C-Terminal, line 503
Please check the WHOLE paper again.
Author Response
CRISPR-Cas9 mediated knockout of SagD gene for overexpression of streptokinase in Streptococcus equisimilis
Armi M. Chaudhari1, Sachin Vyas1, Amrutlal Patel1, Vijai Singh2, Chaitanya G. Joshi1 and Madhvi Joshi1* Author response sheet
Reviewers are appreciated by authors for conducting critical reviews and flagging crucial topics in order to make our work more publishable. Major changes for second revision is highlighted in red and blue color.
Reviewer comments: I thank the authors on comments and improved manuscript. I find this version much better but there are still a lot of typos unfortunately.
Please check that Figures and Tables are always written with the capital letter in the text, genes in italic, proteins with the capital letter, species in italic. Please check lines: 41, 45, 56, 57, 58, 59, 77, 88, 88, and many others ...
Author’s response: In manuscript protein were annotated as SagBCD (COMPLEX), genes where sagD, skc, etc, and organisms’ names were highlighted in italics for example S. equimilius.
Reviewer comments: in materials and methods, also it is SDS-PAGE (line 227
Author’s response: in whole manuscript SDS-page is improved to SDS-PAGE.
Reviewer comments: add "from ribonucleases" in line 250
Author’s response: Thank you for highlighting this missing of from can alter meaning of whole sentence from is added.
Reviewer comments: please define acronym MASCOT
Author’s response: MASCOT is MS-MS search engine suitable citations and acronym is added.
Reviewer comments: in highlights section - sagD gene in italic
Author’s response: SagD gene in italics is added.
Reviewer comments: C-terminal, not C-Terminal, line 503
Author’s response: C-Terminal is improved to C-terminal
Reviewer comments: Please check the WHOLE paper again.
Author’s response: we have amended changes in figures also, and resolution is improved. We have tried to cover all typos.

Reviewer 3 Report
Regarding obtained mutants, as I understood, both of the independently obtained mutated strains were characterised, and presented results are mean values from both strains? If yes, please state clearly in the manuscript or explain undertaken approach.
Please improve in the general language of the manuscript, also paying attention to the below minor comments:
L72: I meant to write as "Golden Gate"
Figure 1. Please improve 1A,1B,1C as the picture looks stretched.
L246: Write instead of E.charpentier and B.Kreikemeyer: "E. Charpentier and B. Kreikemeyer"
L297-L298: The sentence sounds confusing. Please improve that.
L300-L301: Rephrase.
L306: write: "maintenance"
Author Response
CRISPR-Cas9 mediated knockout of SagD gene for overexpression of streptokinase in Streptococcus equisimilis
Armi M. Chaudhari1, Sachin Vyas1, Amrutlal Patel1, Vijai Singh2, Chaitanya G. Joshi1 and Madhvi Joshi1* Author response sheet
Reviewers are appreciated by authors for conducting critical reviews and flagging crucial topics in order to make our work more publishable. Major changes for second revision is highlighted in red and blue color.
Reviewer comments: Regarding obtained mutants, as I understood, both of the independently obtained mutated strains were characterised, and presented results are mean values from both strains? If yes, please state clearly in the manuscript or explain undertaken approach.
Author’s response: In material and methods clarification is added.
Reviewer comments: Please improve in the general language of the manuscript, also paying attention to the below minor comments:
L72: I meant to write as "Golden Gate"
Author’s response: GOLDEN GATE is improved to Golden Gate.
Figure 1. Please improve 1A,1B,1C as the picture looks stretched.
Author’s response Image quality is improved.
L246: Write instead of E.charpentier and B.Kreikemeyer: "E. Charpentier and B. Kreikemeyer"
Author’s response Citations is improved.
L297-L298: The sentence sounds confusing. Please improve that.
Author’s response : Many scientists are concerned about the stability of streptokinase (in terms of protein degradation), for example the Icikinase injection vial of streptokinase, which has a half-life of only 18 minutes.
L300-L301: Rephrase.
Author’s response : Maintaining pH and a stable cation-based buffer system while culturing in growth media has been shown to increase streptokinase production and will be used in our future experiments.
L306: write: "maintenance"
Author’s response: Improved.
Apart from above concerns, In manuscript protein were annotated as SagBCD (COMPLEX), genes where sagD, skc, etc, and organisms’ names were highlighted in italics for example S. equimilius.
